# Renal Cell Carcinoma or Oncocytoma? The Contribution of Diffusion-Weighted Magnetic Resonance Imaging to the Differential Diagnosis of Renal Masses

**DOI:** 10.3390/medicina58020221

**Published:** 2022-02-01

**Authors:** Melike Metin, Hasan Aydın, Mustafa Karaoğlanoğlu

**Affiliations:** 1Radiology Department, Medical Park University, Istanbul 34262, Turkey; melikerusenmetin@gmail.com; 2Radiology Department, Oncology Research Hospital, University of Health Sciences, Ankara 06610, Turkey; 3Radiology Department, Yıldırım Beyazıt University, Ankara 06620, Turkey; mustafakaraoglanoglu@yahoo.com

**Keywords:** ADC, renal, cell, carcinoma, oncocytoma, DWI

## Abstract

*Background and Objectives*: Renal Cell Carcinoma (RCC) accounts for 85% and oncocytomas constitute 3–7% of solid renal masses. Oncocytomas can be confused, especially with hypovascular RCC. The purpose of this research was to evaluate the contribution of diffusion-weighted imaging (DWI) and contrast-enhanced MRI sequences in the differential diagnosis of RCC and oncocytoma *Materials and Methods*: 465 patients with the diagnosis of RCC and 45 patients diagnosed with oncocytoma were retrospectively reviewed between 2009 to 2020. All MRI acquisitions were handled by a 1.5 T device (Achieva, Philips Healthcare, Best, The Netherlands) and all images were evaluated by the consensus of two radiologists with 10–15 years’ experience. The SPSS package program version 15.0 software was used for statistical analysis of the study. Chi-square test, Mann–Whitney U test or the Kruskal–Wallis tests were used in the statistical analysis. A receiver operating characteristic (ROC) curve was used to calculate the cut-off values *Results*: The results were evaluated with a 95% confidence interval and a significance threshold of *p* < 0.05. ADC values (*p* < 0.001) and enhancement index (*p* < 0.01) were significantly lower in the RCC group than the oncocytoma group. *Conclusion*: DWI might become an alternative technique to the contrast-enhanced MRI in patients with contrast agent nephropathy or with a high risk of nephrogenic systemic fibrosis, calculation of CI of the oncocytoma and RCCs in the contrast-enhanced acquisitions would contribute to the differential diagnosis.

## 1. Introduction

In many cases, the differential diagnosis of renal cell carcinoma (RCC) and oncocytoma cannot be made preoperatively by diagnostic imaging methods. Among renal masses, RCC represents approximately 80% of all renal tumors and 75–80% of these are histologically classified as clear RCC [1,2,3], while oncocytomas constitute 3–7% of solid renal masses [4]. Although not observed in all cases, typical imaging findings of oncocytoma are the presence of a central scar, homogeneous contrast distribution, and wash out in the late dynamic venous phases [5,6]. Diagnostic imaging could not clarify the nature of renal masses (oncocytomas vs. RCC) preoperatively. Indeed, oncocytomas could have a similar feature compared to RCC (e.g., oncocytomas with cystic or hemorrhagic changes could resemble hypovascular RCC) [7]. Thus, the differential diagnosis between renal masses is necessarily due to histopathologic pattern [5,6,7].

Recently, diffusion-weighted imaging (DWI) was also included in the differential diagnosis of renal masses in addition to magnetic resonance imaging (MRI) protocols. Based upon the diffusion properties of water molecules, DWI provides the characterization of non-invasive biological tissues via providing information about the biophysical properties of tissues such as cell organization, microstructure and microcirculation. The diffusion coefficient is a measure of mobility at the molecular level and the apparent diffusion coefficient (ADC) map shows the absolute value of the measured diffusion magnitude, which is used in place of the diffusion coefficient [8,9].

The current study aimed to investigate the contribution of DWI and contrast-enhanced MRI series to the differential diagnosis of RCC and oncocytoma.

## 2. Materials and Methods

A total of 465 patients with a diagnosis of RCC and 45 patients diagnosed with oncocytoma were retrospectively reviewed between 2009 and 2020. Patients without MRI (diagnosed only with Multislice computed tomography) or with non-contrast MRI were excluded. As a result, a total of 148 masses (127 RCCs and 21 oncocytomas) were included in the study. All renal masses were sampled by surgical interventions without any biopsy.

All of the RCC and oncocytomas were diagnosed histopathologically by Radical Nephrectomy or nephron-sparing surgeries, including partial nephrectomy and tumor enucleation (127 RCC: 82 with Radical Nephrectomy and 45 with partial nephrectomy) (21 Oncocytomas: 10 with surgical excision by enucleation, 6 with partial nephrectomy and 5 with Radical Nephrectomy).

All MRI acquisitions were performed by a 1.5 T device (Achieva, Philips Healthcare, Best, The Netherlands) with 32 mT. gradient strength unit with a phased-array torso XL coil, and all images were evaluated by the consensus of two radiologists with 10–15 years’ experience. DWI was performed in the transverse plane by using a spin-echo echo-planar imaging sequence (SE-EPI) with fat suppression and breath-hold acquisition. The parameters were: repetition/echo inversion time of 12,000/100/2200 ms, diffusion gradient encoding in 3 orthogonal directions; gradient amplitude (b-value): 0–800 s/mm^2^. The field of view (FOV) was 385 mm, matrix size was 160 × 110 pixels, slice thickness was 6 mm, section gap was 1 mm and the number of excitations (NEX) was 1. DWI scans were acquired before contrast-enhanced T1-weighted imaging, and the acquisition time was about 2.23 min.

ADC values were not statistically different for the most commonly used b-values (maximum of 500 and 600 vs. 800 vs. 1000) on different scanner systems at different institutions). However, there was a trend of decreasing ADC values with increasing b-values. Magnetic resonance diffusion signal and ADC maps were dependent on TR and TE imaging parameters as well as a number of diffusion preparation pulses, but not on the NEX [10,11]. Selective ADC of renal tumors, excluding cystic and necrotic areas, provides better discriminatory ability than whole lesion ADC to differentiate RCC from oncocytomas with higher sensitivity and specificity and also selective ADC excluding cystic and necrotic areas are preferable to whole lesion ADC as an additional tool to multiphasic MRI to differentiate RCC from other renal lesions, especially oncocytomas, whether the highest b-value is 800 or 1000 [10].

At 1.5 T. scanner, we used a relatively long TR, minimum available TE, at least one diffusion preparation pulse, providing sufficient Signal to noise ratio (SNR), b = 800 s/mm^2^ value was acquired and measurements were performed from the solid parts of the tumors to improve the accuracy of ADC values of the renal masses.

Pixel-based ADC maps were reconstructed with a commercially available workstation. ADC values of the detected masses were measured by using an average 20 mm diameter region of interest (ROI) at b = 800 s/mm^2^ The elliptical or rectangular ROIs were placed to the solid-appearing and non-necrotic portions of the renal masses by two experienced Genitourinary radiologists who matched the signal intensity changes on contrast-enhanced T1-weighted imaging as necrotic, cystic and hemorrhagic parts of tumors were very heterogeneous, and tend to mask ADC decrease related to cell proliferation [9,10].

To confirm the reproducibility of the ADC values, multiple measurements were performed, and the averages of these calculations were taken into the final analysis. At final analysis, renal masses were quantitatively analyzed by a local software program with calculation of the ADC values according to the linear regression analysis of the function S = S0 × exp (−b × ADC), where S is the signal intensity after application of the diffusion gradient, and S0 is the signal intensity on the DWI acquired at b = 0 s/mm^2^.

The cut-off value was determined per ADC results of oncocytomas and RCCs; the contrast index (CI) of each mass was determined by calculating the venous phase/arterial phase signal intensity (SI) in the dynamic series.

### Statistical Analysis

The SPSS (Statistical Package for Social Sciences) package program version 25.0 software (SPSS Inc., Chicago, IL, USA) was used for statistical analysis of the data obtained in the study. Descriptive statistics were presented as frequencies, percentages, means, standard deviations, medians and ranges. A comparison of the categorical variables between the groups were handled with the Chi-square test, while numerical variables were compared with the Mann–Whitney U test or the Kruskal–Wallis test. Receiver operating characteristic (ROC) analysis was used to calculate the cut-off value. The results were evaluated with a 95% confidence interval and a significance threshold of *p* < 0.05.

## 3. Results

A total of 148 patients (127 RCC and 21 oncocytoma patients) were included in this research; 70.9% were male and 29.1% were female in the RCC group, whereas 76.2% of the oncocytoma cases were male and 23.8% were female (*p* > 0.05). None of the patients had presurgical therapies and extra-renal tumor history of the patients was not analyzed in the anamnesis. The ages ranged from 21 to 84 in the RCC group, with a mean of 59.47 ± 12.15 years. On the other hand, the mean age was 64.29 ± 14.71 years (36–78) in the oncocytoma group. There was no significant difference between two the groups related to the age (*p* > 0.05). None of the patients had a multifocal tumor; only the patients with solid masses in his/her right or left kidney were involved in the study. The localization of the masses was higher in the left kidney in both groups, and there was no statistically significant difference between two groups concerning the mass localization (*p* > 0.05). The size of tumor lesions varied between 12 and 130 mm in the RCC group, mean 71 ± 21 mm and 26 to 104 mm in the oncocytoma group, mean 65 ± 23 mm. Although the mass diameter was somewhat larger in the RCC group, there was no statistically significant difference between two groups (*p* > 0.05).

However, the ADC value (*p* < 0.001) and enhancement index (*p* ˂ 0.01) were significantly lower in the RCC group than the oncocytoma group in the statistics (Table 1 and Table 2).

Throughout the 127 RCC masses, 89 (70.1%) were clear cell (CC) RCC, 9 (7.1%) were chromophobe cell (CRH) RCC, and 29 (22.8%) were papillary type (P) RCC. When these three subtypes were compared to each other, concerned with the ADC values, there were no statistically significant difference among subtypes (*p* > 0.05) (Table 3).

The area under ROC curve was 0.753 ± 0.055 (95% CI: 0.646–0.860) (*p* ˂ 0.001) in the differentiation of RCC and oncocytoma (Figure 1). The best cut-off point was determined as 1.150. The sensitivity, specificity, positive predictive value (PPV) and negative predictive values (NPV) of the ADC value at this cut-off range, were calculated as 59.1, 85.7, 96.2 and 25.7%. On the other hand, cut-off point for the enhancement index was presented as 1.520 for the differentiation of RCC and oncocytoma, sensitivity–specificity-PPV-NPV of the CI at this cut-off range were about 71, 60, 93.3 and 19.8%, respectively. The combination of ADC value and CI in the ROC curve had revealed 95% sensitivity and 75% specificity in the differential diagnosis of RCC and oncocytomas (Figure 2).

## 4. Discussion

Today, 10–30% of benign renal masses were incidentally detected in the histopathological examinations of patients who had undergone resections [12]. Therefore, benign lesions had to be precisely characterized preoperatively in order to prevent unnecessary operations. According to Ultrasonography (USG) data, the incidence of malignant renal tumors was increasing per year consecutively [5,12]. Mass lesions could be detected incidentally in asymptomatic patients with more frequently applied imaging methods as a leading cause for this [3]. Partial nephrectomy was preferred in younger patients, whereas focal ablation methods were preferred in elderly patients with comorbidities [12,13].

The principal imaging methods used in the diagnosis of renal tumors were USG, computed tomography (CT) and MRI. However, it was difficult to make a sufficient differentiation between benign and malignant tumors with those techniques. Thus, additional imaging modalities should be required for a differential diagnosis, as biopsy was still the preferred diagnostic method [12]. However, since biopsy was an invasive method, a reliable non-invasive technique could reduce unnecessary surgical interventions and could provide appropriate treatment options for patients [3,13].

RCCs were divided into different subtypes histopathologically; each subtype had different metastatic potentials, different behaviors and variable survival rates. The most common subtypes were clear cell RCC (70–80%); others were papillary RCC (10–15%) and chromophobe RCC (5%) [14,15] (Figure 3a,b).

Oncocytomas are benign renal tumors. They should be differentiated from RCCs preoperatively to determine the exact treatment method and estimate the survival [6]. There are many studies which were performed by using imaging methods to differentiate RCC and oncocytoma. Oncocytomas were generally benign, well-circumscribed, solid tumors that could contain central scars and were homogeneously contrast enhanced [16,17]. However, they might show changes such as necrosis and bleeding. Therefore, typical imaging features should not be present, and differential diagnosis with RCC may not be possible [4,5,6,18]. Moreover, even if the central scar was characteristic for oncocytoma (45%), it was not a diagnostic tool; a central scar could also be seen in some RCC types [5,6]. On the other hand, necrotic areas that could be encountered in RCCs, might mimic the central scar [18]. Hence, new information and research were needed to concern the differential diagnosis of solid renal masses with imaging methods *(*Figure 4a–c) In our study, 127 RCCs and 21 oncocytomas were evaluated by ADC measurements for the differential diagnosis of RCC, RCC subtypes and oncocytoma. Additionally, the enhancement patterns, which contributed to the differential diagnosis by calculating the CI of both types of tumors, were also investigated.

DWI was an advanced high MRI technique based on the molecular mobility of the water [19]. In DWI, the mobility of water molecules was limited in lesions with high cellularity, causing a more hyperintense appearance [20]. On the other hand, most of the malignancies were hypointense compared with the surrounding tissues due to higher cellularity in ADC mapping. Measurable data obtained with DWI could give an idea about the nature of the lesions in various parts of the body and helped to characterize them without using any intravenous contrast agents [21]. Recent research based on the MRI of renal tumors has shown that ADC values obtained by DWI could provide new quantitative data to distinguish benign tumors from malignant ones, and consequently might help to identify pathological subtypes of neoplasms [22,23].

In our sample, the ADC value was significantly lower in the RCC group than the oncocytoma group, statistically, with the best cut-off value of 1.150. The sensitivity and specificity of the ADC at this value was 59.1 and 85.7%, respectively. Taouli B et al. [22] concluded that DWI could provide additional information to contrast enhanced studies in the diagnosis of oncocytoma and histologic subtypes of RCC that was consistent with our findings. Anna K. et al. [24] stated that they included 26 oncocytomas, 97 ccRCC and 29 pRCC patients, and found a significant statistical difference between the papillary type RRC and clear cell RCC, but they did not find a significant difference between clear cell RCC and oncocytoma. However, in our study, necrotic areas were not included in ADC measurement, and there was a significant difference between RCC and oncocytomas concerning the ADC values.

In their research, Kim et al. [25] stated that the enhancement pattern of oncocytomas in the nephrogram and early pyelogram phase in the form of segmental inverse patterns was pathognomonic. However, studies examining the differentiation of RCC from oncocytoma based on the nephrogram phase presented inconsistent results [26,27]. In our patients, venous phase SI/arterial phase SI was calculated as CI for both groups, which was significantly lower in the RCC group than in the oncocytoma group. In dynamic MRI examinations, lesions with high cellularity hold more contrast in the arterial phase, and this information was consistent with our results (Figure 5a,b). Further studies were present in the literature that had supported our results. Gakis et al. [26] reported that oncocytomas hold more contrast than RCCs in the nephrogram phase. Young et al. [28] compared clear-cell RCCs and oncocytomas with regard to contrast enhancement and evaluated the difference in contrast enhancement between the kidney mass and normal renal cortex. They reported a significant difference in relative corticomedullary signal intensity between oncocytomas and clear cell RCCs.

By calculating ADC values and SI index of 26 oncocytomas and 16 chromophobe RCCs (tumor-to-spleen SI ratio), Chloe et al. [29] found significantly higher ADC values in the oncocytomas. They showed rapid contrast accumulation in oncocytomas compared to the RCCs. These findings were consistent with our study (Figure 6a–c).

Basara et al. [30] found a significant difference between oncocytomas and chromophobe RCCs concerned with the contrast enhancement and stated that oncocytomas had more contrast accumulation in all phases than chromophobe RCCs. We also reached similar results in our statistical analysis without categorizing RCCs into subtypes.

Angin et al. [31] conducted an ADC histogram analysis of lesions, and they included renal masses of less than 4 cm in size. They found a significant statistical difference between clear cell RCC and papillary RCC, chromophobe RCC and fat-containing angiomyolipomas (AMLs), but no difference was reported between oncocytomas and other RCC types and/or low-fat containing AMLs. They claimed that ADC calculations might be affected from measurements through the cystic areas that might exist in large lesions for the masses greater than 4 cm. However, in our study, we cared about taking ADC measurements from the solid components of the lesions.

Many studies were conducted regarding the differential diagnosis of RCC subtypes by using imaging methods. Many of those research studies assessed ADC measurements of the renal masses using DWI. In a study performed by Andreas et al. [32] with a total of 117 patients, including all renal tumor types, they reported that clear-cell RCC was significantly more prevalent than other RCC subtypes (chromophobe, papillary and unclassified RCC). However, in the same study, it was stated that there were some overlaps between ADC values of the clear cell RCC subtype and other renal tumors. In our study, no significant difference was found between RCC subgroups regarding with the ADC values and CIs (*p* > 0.05). Additionally, there were similar studies in the literature that had supported our results [23,33].

The clinical implication of this research was to supply adequate information for the differential diagnosis of RCC and oncocytomas by DWI-ADC value measurements and to decrease the necessity of surgical interventions to the lowest degree for the proper diagnosis of those renal masses. By the improvement and advancing high DWI techniques such as Diffusion Kurtosis imaging, Diffusion Tensor imaging, high-ultra high b-value DW images, etc., unnecessary surgical approaches would be totally avoided for the diagnosis of renal masses [34,35]. Diagnostic MRI and DWI-ADC construction could potentially aid in the early diagnosis of renal masses and influence the differential diagnosis of RCC and oncocytomas non-invasively, leading to increased survival outcomes that were adjusted by the diagnosis of RCC in the early stages with short sizes limited to the kidneys without any metastasis and overcome and/or distinctly decreased the rate of surgical procedures by distinguishing oncocytomas from RCC subtypes [4,5,6,23,31,33].

## 5. Limitations

This research was a retrospectively designed study and since most of the RCC patients (70.1%) were in the clear cell carcinoma subtype, no appropriate comparison could be made between the ADC values of the subtypes and the ADC values of oncocytomas. It was necessary to examine whether there were differences in ADC values between other subtypes of RCC (such as papillary cell carcinoma and chromophobe cell carcinoma) and oncocytomas by using larger sampling sizes.

## 6. Conclusions

This study aimed to investigate whether there was a significant difference in the ADC and CI values of RCCs and oncocytomas by using the dynamic contrast-enhanced MRI and DWI methods. By using the estimated cut-off value (1.150), MRI could have a significant contribution to the differential diagnosis between RCC and oncocytoma. These results were obtained by using an average 20 mm ROI in diameter. In addition, CI was significantly different between both groups. We had considered that Acquisition of CI of RCCs as 1.378 and CI of oncocytomas as 2.200 in MRI evaluations should contribute to the differential diagnosis of RCC and oncocytomas. According to the yields of this research, DWI might become an alternative modality to the contrast-enhanced MRI in patients with contrast agent nephropathy or with a high risk of nephrogenic systemic fibrosis. Additionally, we thought that the calculation of the CI of the oncocytoma and RCCs in the contrast-enhanced examinations would precisely contribute to the differential diagnosis. Measured ADC and CI of the oncocytoma and RCCs with cut-off values and combined use of them, which had supplied statistically significant difference between two groups with 95% sensitivity in the differential diagnosis of both groups, were the innovation of this research, leading to a concrete difference from similar publications in the literature. There was no significant difference between RCC subtypes concerning the ADC values and CI.

## Figures and Tables

**Figure 1 medicina-58-00221-f001:**
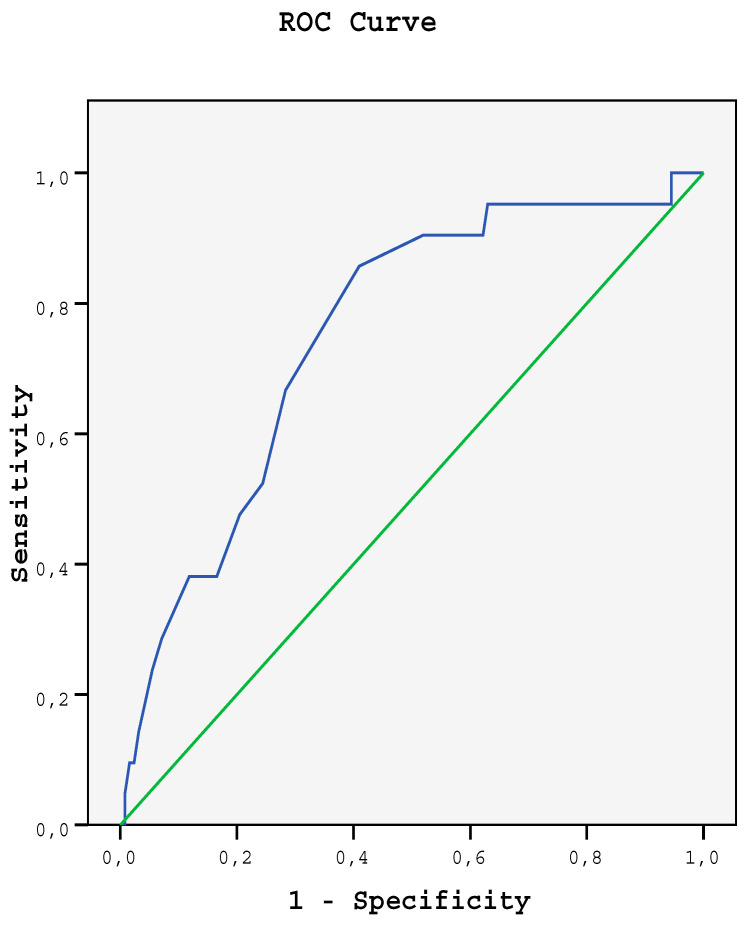
ROC curve of the ADC values for the differentiation of RCC and oncocytoma separation.

**Figure 2 medicina-58-00221-f002:**
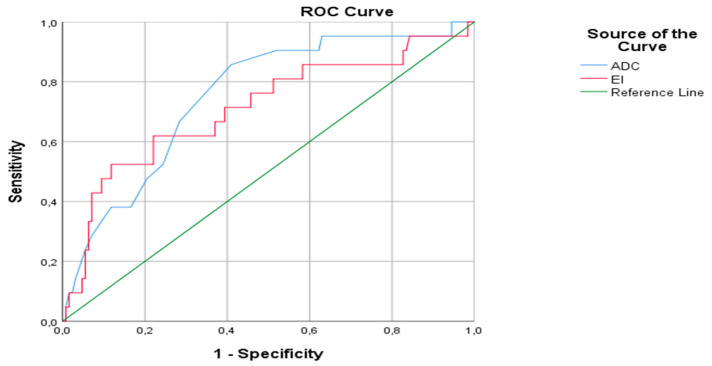
ROC curve of the combined ADC value and CI for the differentiation of RCC and oncocytoma.

**Figure 3 medicina-58-00221-f003:**
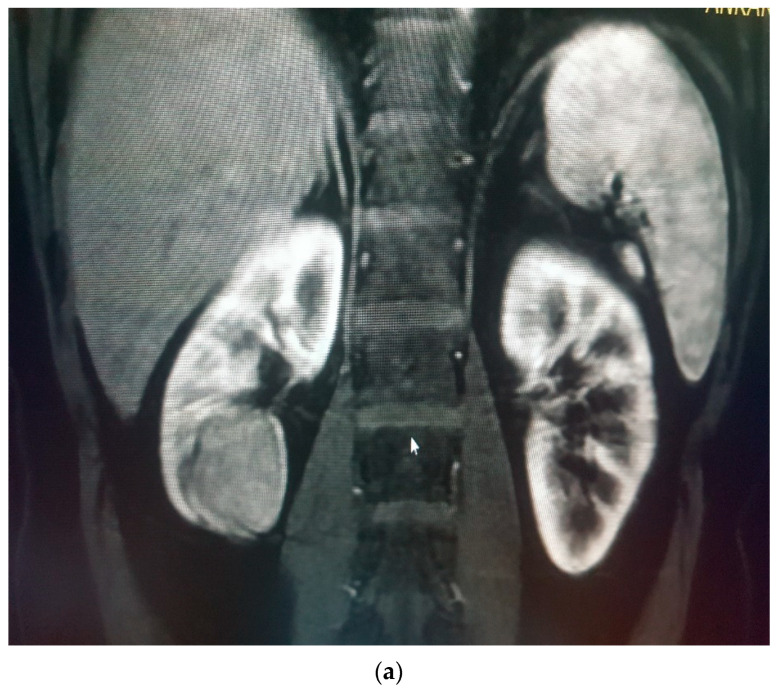
(**a**) Renal mass (RCC) in the lower pole of right kidney, low enhancing proportional to the renal cortex but very dark with low ADC and (**b**) concrete restricted diffusion in diffusion-weighted imaging (DWI).

**Figure 4 medicina-58-00221-f004:**
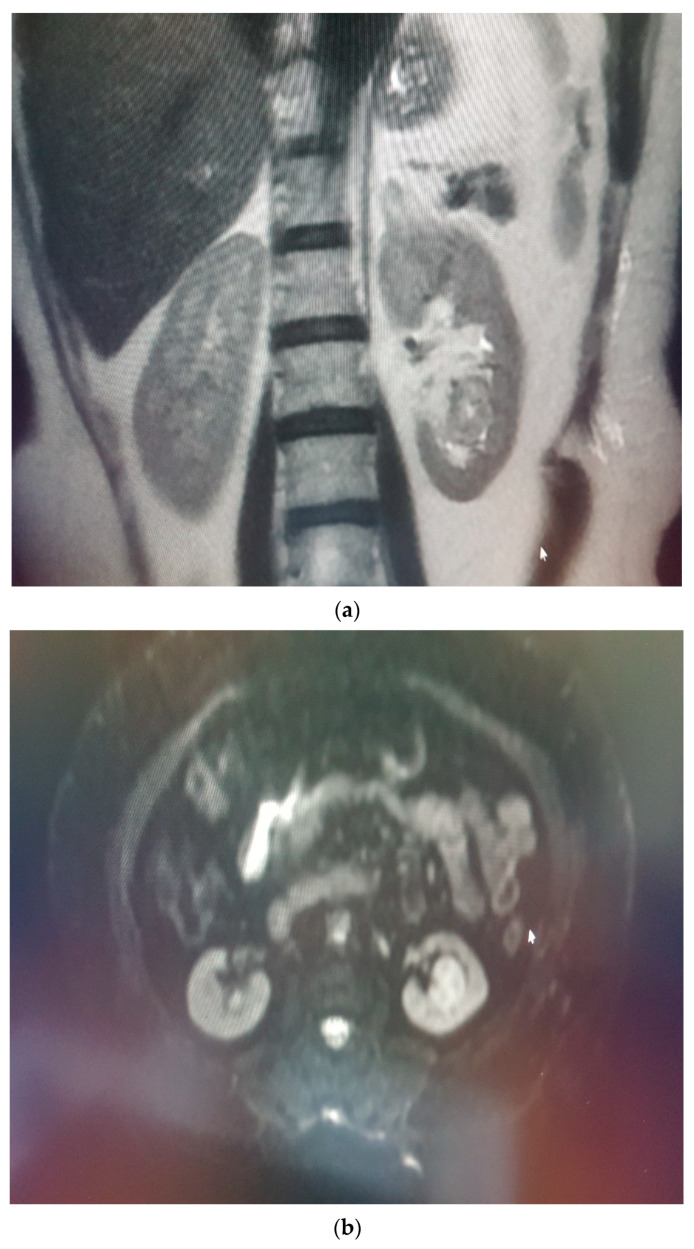
(**a**) Oncocytoma in the the lower pole of left kidney, (**b**) iso-mild hyperintense in T2W images with obvious restricted diffusion and (**c**) hyperintense in DWI.

**Figure 5 medicina-58-00221-f005:**
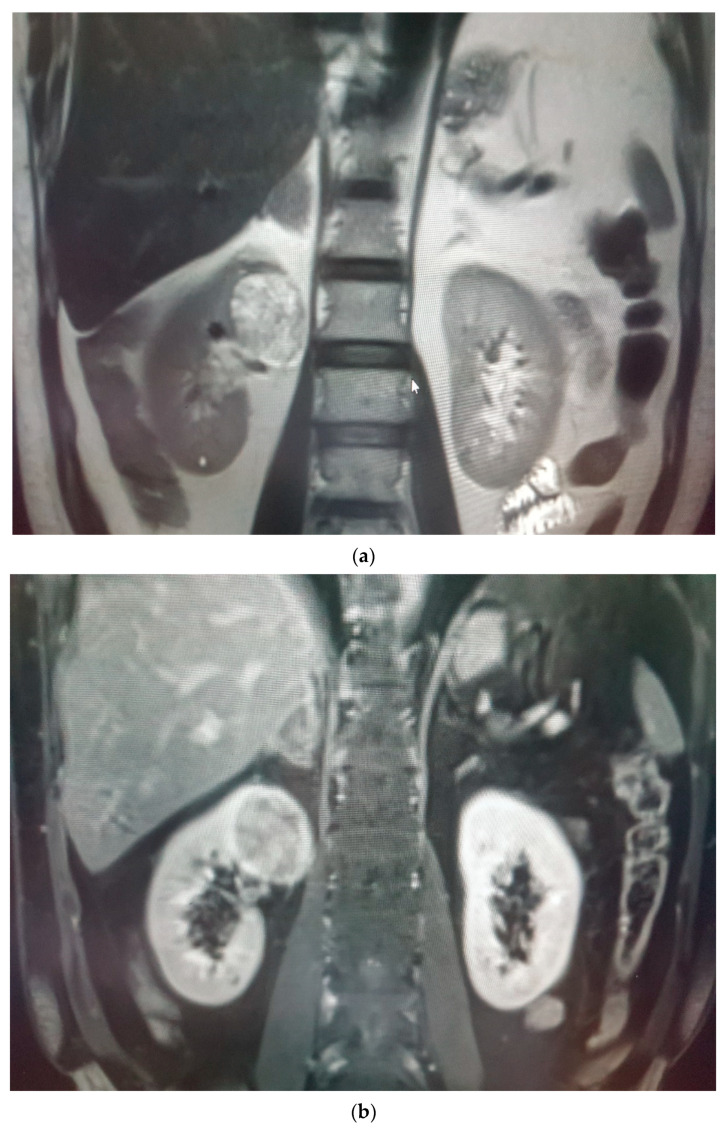
(**a**) Renal mass lesion (RCC) in the upper pole of right kidney, (**b**) hyperintense in T2W images with high contrast enhancing.

**Figure 6 medicina-58-00221-f006:**
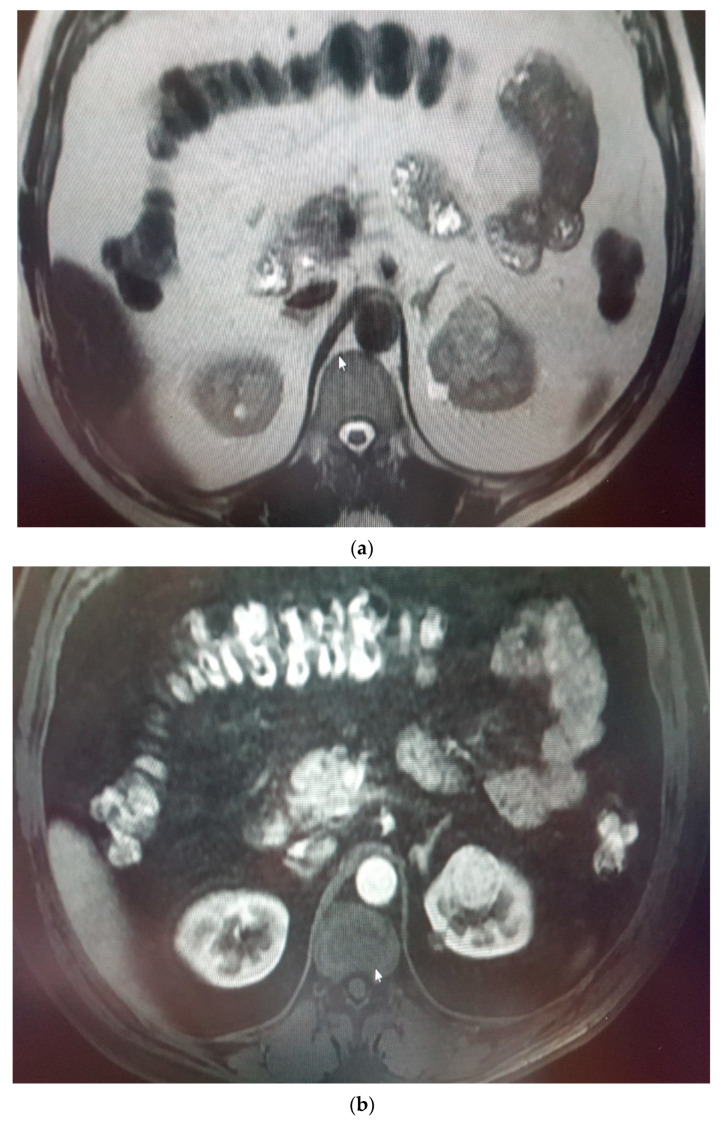
(**a**) Oncocytoma in the upper pole of left kidney, (**b**) iso-hyperintense in T2W images with high diffusion restriction and (**c**) moderate contrast enhancing.

**Table 1 medicina-58-00221-t001:** Comparison of RCC and oncocytoma groups with regard to the gender and localization.

	RCC		Oncocytoma		*p*
	*N*	%	*n*	%	
Sex					0.616
Male	90	70.9	16	76.2	
Female	37	29.1	5	23.8	
Localization					0.211
Right	61	48.0	7	33.3	
Left	66	52.0	14	66.7	

RCC: renal cell carcinoma.

**Table 2 medicina-58-00221-t002:** Comparison of age, diameter, ADC and enhancement index between RCC and oncocytoma groups.

	RCC				Oncocytoma				*p*
	Mean	SD	Median	Range	Mean	SD	Median	Range	
Age	59.47	12.15	61.00	21.00–84.00	64.29	14.71	71.00	36.00–87.00	0.070
Diameter of the mass	49.07	27.86	40.00	12.00–130.00	39.05	19.60	37.00	15.00–90.00	0.135
ADC	1.15	0.73	1.10	0.18–7.70	1.55	0.53	1.40	0.50–2.80	<0.001
Contrasting index	1.520	0.551	1.378	0.867–3.413	2.112	0.799	2.200	0.900–3.870	0.001

SD: Standard deviation; ADC: apparent diffusion coefficient.

**Table 3 medicina-58-00221-t003:** Distribution of ADC values per the RCC subtypes.

	Mean	SD	Median	Range	*p*
Clear cell	1.241	0.818	1.200	0.18–7.70	0.112
Chromophobe	0.929	0.296	0.900	0.49–1.50	
Papillary	0.950	0.447	0.800	0.22–2.40	

SD: Standard deviation.

## Data Availability

All available datas were reported for the research.

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
