# Peer review of "Renal Cell Carcinoma or Oncocytoma? The Contribution of Diffusion-Weighted Magnetic Resonance Imaging to the Differential Diagnosis of Renal Masses"

_medicina, 2022, doi:10.3390/medicina58020221_

Round 1

Reviewer 1 Report

The search had evaluated the contribution of DWI and contrast-enhanced MRI sequences in the differential diagnosis of RCC and oncocytoma. However, I have several concerns to this manuscript:

1.There are several similar studies in publication. What is the innovation in your study?

2.As we known, the DWI was influenced by many factors. What measures do you use to improve the accuracy of ADC value?  

  1. How to construct the ADC maps? Please state in detail.
  2. The study showed that the ADC values and enhancement index were statistically significantly lower in the RCC group than the oncocytoma group. How about the ROC curve of ADC value combine to the enhancement index for the differential diagnosis?
  3. Some spelling and grammar check needed for completion of the manuscript.

Reviewer 2 Report

Among renal masses, renal cell carcinoma (RCC) approximately represents the 80% of all renal tumors and 75-80% of these are histologically classified as clear RCC. Diagnostic imaging could not clarify the nature of renal masses (oncocytomas vs RCC), preoperatively. Indeed, oncocytomas could have similar feature compared to RCC (e.g oncocytomas with cystic or hemorrhagic changes could resemble hypovascular RCC). Thus, the differential diagnosis between renal masses is necessarily due to histopathologic pattern. Recently, diffusion-weighted imaging (DWI) is also included in the differential diagnosis of renal masses in addiction to Magnetic resonance imaging (MRI) protocols. The current study aimed to investigate the contribution of DWI and contrast-enhanced MRI series to the differential diagnosis of RCC and oncocytoma.

Comments to Authors

Authors should be congratulated for the great work. The topic is interesting and challenging and it offer a new diagnostic perspective for patients with suspicious renal masses. A moderate revision of English is required, the methodology is robust, and tables and figures are clear. The manuscript presents several points to clarify:

  1. Authors should enlighten the baseline characteristics of patients enrolled (e.g the renal samples origin, the surgical approach [if radical or not], size of lesion, the multifocality of tumor, pre-surgical therapies, or patients’ tumor history).
  2. What is the clinical implication of the study? Could the surgical intervention be avoided? Which survival outcomes were influenced by the imaging differential diagnosis?

Round 2

Reviewer 1 Report

The revised manuscript is good. 1. Some spellings were further needed to check for completion of the manuscript. For example:“Size of the renal tumor lesions vary between 12-130 mm in the RCC group, mean 71+- 21 and 26-104 mm in the oncocytoma group, mean 65+- 23.” 2.Minor question: There are 10 cases with oncocytoma were treated with enucleation, but all renal masses were sampled by surgical interventions without any biopsy. How to get the pathological diagnosis?

Author Response

1-Spelling errors were corrected.

2-Pathological diagnosis for the oncocytomas were explained.

Reviewer 2 Report

Authors should be congratulated for the great work. The topic is interesting and challenging and it offer a new diagnostic perspective for patients with suspicious renal masses. The methodology is robust, and tables and figures are clear. Authors had exhaustively answered to my sentences and points of view. The manuscript is a suitable for a publication.

Author Response

1-A brief check for English spelling was performed.